# The Non-Thermal Radio Emissions of the Solar Transition Region and the Proposal of an Observational Regime

Baolin Tan [1,2,3,*], Jing Huang [1,2,3], Yin Zhang [1,3], Yuanyong Deng [1,3], Linjie Chen [3], Fei Liu [1,3], Jin Fan [1,3] and Jun Shi [3]

- [1] National Astronomical Observatories of Chinese Academy of Sciences, Beijing 100012, China; huangj@nao.cas.cn (J.H.)
- [2] School of Astronomy and Space Science, University of Chinese Academy of Sciences, Beijing 100049, China
- [3] Key Laboratory of Solar Activity and Space Weather, National Space Science Center, Chinese Academy of Sciences, Beijing 100190, China; shijun@nssc.ac.cn (J.S.)
- [*] Correspondence: bltan@nao.cas.cn

**Abstract:** The transition region is a very thin but most peculiar layer in the solar atmosphere located between the solar chromosphere and the corona. It is a key region for understanding coronal heating, solar eruption triggers, and the origin of solar winds. Here, almost all physical parameters (density, temperature, and magnetic fields) have the maximum gradient. Therefore, this region should be highly dynamic, including fast energy releasing and transporting, plasma heating, and particle accelerating. The physical processes can be categorized into two classes: thermal and non-thermal processes. Thermal processes can be observed at ultraviolet (UV) and extreme ultraviolet (EUV) wavelengths via multi-wavelength images. Non-thermal processes accelerate non-thermal electrons and produce radio emissions via the gyrosynchrotron mechanism resulting from the interaction between the non-thermal electrons and magnetic fields. The frequency range spans from several GHz to beyond 100 GHz, in great number of bursts with narrowband, millisecond lifetime, rapid frequency drifting rates, and being referred to as transition region small-scale microwave bursts (TR-SMBs). This work proposes a new type of Solar Ultra-wide Broadband Millimeter-wave Spectrometer (SUBMS) that can be used to observe TR-SMBs. From SUBMS observations, we can derive rich dynamic information about the transition region, such as information about non-thermal energy release and propagation, the flows of plasma and energetic particles, the magnetic fields and their variations, the generation and transportation of various waves, and the formation and evolution of the source regions of solar eruptions. Such an instrument can actually detect the non-thermal signals in the transition region during no flare as well as the eruptive high-energy processes during solar flares.

**Keywords:** solar transition region; radio emission; magnetic field; gradient





## 1. Introduction: Why Focus on the Solar Transition Region?

In modern solar/stellar physics, we face a series of long-standing scientific challenges, including the mystery of coronal heating; the triggering mechanism of powerful eruptions, such as solar/stellar flares and coronal mass ejections (CMEs); and the origin of solar/stellar winds. In order to answer these problems, we have to track and study how massive amounts of matter and energy are transmitted and released from the solar/stellar interior into the upper atmosphere. Here, we encounter a very unique atmospheric layer—the transition region. Take the Sun as an example. Figure 1 shows that the solar transition region is a very thin layer with a thickness of only a few hundred km and a height of a few thousand km (from 0.14 Mm to 5.7 Mm, depending on observations at different wavelengths [1]); it sits above the solar photospheric surface and is accompanied by complex motions [1,2]. It separates the colder, partially ionized, and frequently collisional chromosphere from the very hot, fully ionized, and collisionless corona. As we know, the temperature changes slowly in the solar photosphere, chromosphere, or corona. But in the thin transition region,

the temperature quickly increases from about $2 \times 10^4$ K at the top of the chromosphere to near one million K at the base of the corona, causing rapid changes in plasma density, magnetic fields, and other physical parameters [3,4]. Therefore, it is not a simple thermal equilibrium layered structure but a rapidly changing and complex dynamic region with highly heterogeneous magnetic fields and plasmas [5]. The transition region is so unique due to its high gradients of physical parameters. Its formation should be related not only to the process of mass and energy input from the Sun's interior but also to the heat conduction downwards from the upper hot corona. However, it is still not very clear how the transition region is formed, and we need to conduct in-depth research on the energy releasing mechanism and transportation in this region.

We know that the mystery of coronal heating is one of the eight major challenges in contemporary astronomy [6,7]. The existing heating mechanisms can be roughly divided into three categories: wave heating [8,9], reconnection heating [10,11], and magnetic field gradient pumping heating [12,13]. However, so far, there is no consensus on which mechanism truly underlies coronal heating [14]. It is particularly important that all mass and energy required for heating the corona and even driving solar eruptions and the solar wind need to be transported upwards through this thin layer. Therefore, studying the transition region is crucial for solving the three above-stated major problems in the field of solar physics—coronal heating, eruption triggering, and the origin of solar wind.

Observations show that there are many small-scale transient activities which may affect energy release and mass transfer in the solar transition region, such as spicules, small-scale jets, various small bright points, and even parts of nanoflares [15]. Behind these phenomena are both thermal and non-thermal processes. So far, many solar telescopes have been designed specifically for observing the solar transition region at ultraviolet (UV) and extreme ultraviolet (EUV) wavelengths, such as SUMER on SOHO [16], TRACE [17], and IRIS [18]. These observations have provided a lot of information on the transition region, including the morphology and dynamics of spicules, small-scale jets, bright points, and networks [19,20]. These are intrinsically thermal processes. However, if we want to fully reveal the physical essence of the transition region, in addition to carefully observing various thermal processes, observing non-thermal processes is also most important. Here, the non-thermal processes include the rapid energy release by magnetic reconnection and the associated particle accelerations and propagations. All non-thermal processes are associated with the generation and propagation of energetic particles which interact with plasmas and magnetic fields to produce radio emissions. Therefore, radio observations will also play an extremely important role in detecting non-thermal processes in the transition region. As we will discuss in Section 2, the non-thermal electrons will generate centimeter–millimeter-wave emission in the transition region mainly by the gyrosynchrotron mechanism. Considering the magnetic field, plasma density, and temperature, the emission is optically thin, and the spectrum is directly related to the energy distribution of the electrons.

As we know, the frequency of solar radio emissions spans from sub-millimeter waves to beyond kilometer waves. Regardless of the emission mechanism and propagation effect [21], generally, the shorter the wavelength, the closer the source region of the radio emission is to the photosphere; the longer the wavelength, the farther away the source region of the radio emission is from the photosphere. For example, flare-associate radio emissions at centimeter and decimeter wavelengths are produced in the lower corona near the source regions of solar flares, while the radio emissions at meter and decameter wavelengths are generated in the higher corona and always related to the propagation of non-thermal electron beams, CMEs, and the related shock waves [22]. Then, radio emissions in the solar transition region with shorter wavelengths should fall into the centimeter–millimeter wave band.

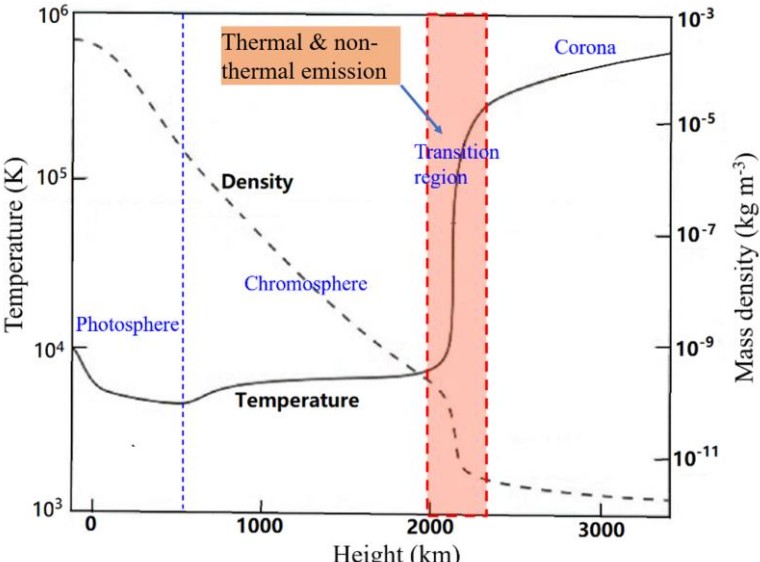

**Figure 1.** The position and the variation characteristics of the temperature and mass density of the solar transition region (plotted from the results of Vernazza et al., 1981 [23]).

The remainder of this work is structured as follows: Section 2 discusses the radio signals of the non-thermal processes in the solar transition region. Section 3 discusses the limitations of radio observations of the transition region. In Section 4, we propose a new Solar Ultra-wide Broadband Millimeter-wave Spectrometer (SUBMS) for observing the signals of the non-thermal processes in the transition region. Section 5 presents the conclusions that can be derived from this work.

## 2. Non-Thermal Radio Emissions from the Solar Transition Region

In the solar transition region, the plasma density rapidly decreases from about $10^{18}$ m$^{-3}$ to $10^{16}$ m$^{-3}$, and the temperature quickly increases from about $2 \times 10^4$ K at the top of the chromosphere to around $4 \times 10^5$ K at the base of the corona [23], and the corresponding plasma frequency ranges from several GHz to around tens of GHz, which is the cutoff frequency for electromagnetic waves that can propagate from the transition region. The frequency of radio emissions coming from the solar transition region must be higher than the above plasma frequency, generally in the centimeter–millimeter-wave frequencies of tens of GHz or above, and their formation height is generally determined by the opacity [24], which depends on the emission mechanism [25]. We know that some powerful flare eruptions can also generate enhanced radio bursts at centimeter–millimeter wavelengths. So, what is the difference between flare-associated centimeter–millimeter-wave radio bursts and centimeter–millimeter-wave emissions from the solar transition region?

Flare-associated centimeter–millimeter-wave bursts are generally produced in fast-varying structures and in the coronal flaring source region. De Jager et al. (1987) [26] and Kaufmann et al. (1986) [27] investigated millimeter-wave bursts with fast quasi-periodic pulsating structures and a pulsating period of about 60 ms in the impulsive phase of a solar flare. These millimeter-wave bursts are attributed to gyrosynchrotron emission by non-thermal electrons and even synchrotron emission by ultra-relativistic electrons in a relatively strong magnetic field [28]. Accompanying these millimeter-wave bursts, almost identical hard X-ray temporal structures have been simultaneously observed. For example, Kaufmann et al. reported that intense and fast-pulsed structures at 90 GHz and relatively smaller amplitude structures at 30 GHz were almost identical to the temporal structures of hard X-ray bursts at an energy range of 24–108 keV at 13:26 UT on 21 May 1984. At the same time, there was almost no enhancement at 7 GHz corresponding to the hard X-ray bursts during the same period [29]. Actually, this event occurred in the preflare phase just 2 min before the onset of a C2.5 flare (a GOES soft X-ray observation shows that the flare

started at 13:28 UT, peaked at 13:31 UT, and ended at 13:55 UT). Further studies indicate that flare-associated millimeter-wave emissions best correlate with hard X-ray emissions above 500 keV [30].

Then, what is the centimeter–millimeter-wave emission in the solar transition region like? We know that compared to the most frequently flaring source region located in the corona, the solar transition region has a higher density (from $10^{16}$ m$^{-3}$ to $10^{18}$ m$^{-3}$) (see Figure 1), a stronger magnetic field (from about 100 G to nearly 2000 G; from the quiet solar region networks to the sunspot region), and lower temperature, ranging from $2 \times 10^4$ K to $4 \times 10^5$ K (also see Figure 1). With these parameters, we may calculate the plasmas β in the range from $10^{-5}$ around the top of the transition region (above the active sunspot regions) to 0.01 near the bottom of the transition region (above the quiet solar region). That is to say, the plasmas still have β ≪ 1. In fact, except for the transition regions above some areas where the magnetic field is too weak (for example, B < 100 G), the solar transition regions above the active sunspot regions, consisting of network structure granules, basically meet the conditions of collisionless magnetized plasma. Such conditions are conducive to thermal bremsstrahlung emission and cyclotron emission by thermal electrons. Obviously, the thermal bremsstrahlung and cyclotron emissions in the solar transition region should be in broadband continuum, which constitutes the fundamental background component of radio emission in the solar transition region. Now that the solar transition region is highly dynamic, there should be frequently non-thermal energy release and conversion occurring here, resulting in the generation of non-thermal electrons. What emission signals will be generated by the interaction between these non-thermal electrons, magnetic fields, and the transition region plasma?

### 2.1. Non-Thermal Electrons in the Transition Region

Let us analyze how non-thermal energy is released in the solar transition region and how non-thermal electrons are generated?

Firstly, the high gradient of temperature and density in the solar transition region inevitably drives the strong convections of the plasma and rapid variations in the magnetic field. The strong convections will produce strong turbulence, and the magnetic field rapid variations may produce various small-scale magnetic reconnections. Both of strong turbulence and magnetic reconnection can accelerate electrons to high energy effectively and release non-thermal energy [31,32].

As for the turbulence acceleration, Li et al. [33] applied the MHD simulation to investigate the electron acceleration based on turbulent current sheets and found that the electrons can be accelerated to more than several hundreds of keV. The turbulent processes in the transition region occur universally anytime and anywhere, whether in the quiet Sun or the active solar region. At the same time, the moving direction of the non-thermal particles generated by turbulent acceleration in the transition region is almost randomly isotropic, so these non-thermal particles are approximately equivalent to a non-thermal component superimposed on the background of the thermal plasma.

Magnetic reconnections should be another important candidate for particle accelerations in the solar transition region [34]. For example, based on observations of several rocket flights, Porter & Dere (1991) reported many small-scale and short-lived (typically < 1 min) jets and turbulent events with non-Gaussian enhancements in both the red and blue wings of transition region lines covering temperatures ranging from $2 \times 10^4$ K to $2 \times 10^5$ K [15]. Although many people think that small-scale explosive phenomena such as micro-flares, nano-flares, spicules, bright points, and even small-scale jets mainly occur in the solar chromosphere (hence why they are called chromospheric fine structures), recent UV observations from the IRIS satellite show that most of them have a temperature of at least $10^5$ K and constitute important elements of the transition region structures. They are associated with magnetic reconnection [15,35]. As we mentioned above, except for some regions with very weak magnetic fields (such as B < 100 G), most of the solar transition region basically meets the conditions of collisionless magnetized plasma, with β < 0.01.

Therefore, magnetic reconnection will be triggered and accelerate electrons [34]. We know that the physical essence of magnetic reconnection particle acceleration is conducted by the induced electric field formed in the magnetic reconnection site, and the current sheet is its main manifestation. The energy of accelerated reconnection electrons can be estimated as follows [36]:

$$E_{acc} \approx 1.23 \times 10^{13} \frac{B_c^2}{n_e} \text{ (keV)} \tag{1}$$

Here, $n_e$ is the electron density (m$^{-3}$) in the transition region, and $B_c$ is the magnetic field strength (G) near the reconnection site. For $B_c \sim 100$ G, $n_e \sim 10^{17}$ m$^{-3}$, we have $E_{acc} \sim 1.2$ keV; when $B_c \sim 500$ G, $E_{acc} \sim 30$ keV, and when $B_c \sim 2000$ G, we will have $E_{acc} \sim 480$ keV. Near the upper part of the transition region, we may assume $n_e \sim 10^{16}$ m$^{-3}$, and when $B_c \sim 500$ G, $E_{acc} \sim 300$ keV; when $B_c \sim 1000$ G, $E_{acc} \sim 1.2$ MeV, which are relativistic electrons. That is to say, $E_{acc}$ may range from several keV to MeV.

Supposing that all electrons in a small-scale current-sheet can be accelerated by magnetic reconnection, the length and width of the small-scale current-sheet are 100 km, respectively, thickness is at scale of 0.1 km, and $n_e \sim 10^{16} - 10^{18}$ m$^{-3}$, then the total number of accelerated electrons can be estimated as $10^{28} - 10^{30}$ in a reconnected current sheet. Such a group of electrons may carry a kinetic energy of about $10^{13} \sim 10^{16}$ J. As the reconnected current sheets may be ubiquitous and randomly distributed in the solar transition region, each current sheet may generate a group of accelerated electrons, and many current sheets will produce many groups of accelerated electrons. If we assume that it is these accelerating particles (ions and electrons) that predominantly heat the corona, we can estimate that there may be a total of $10^5 \sim 10^8$ active current sheets per second (this number is interestingly close to the number of type II spicules [37]) and a total of $10^{34} \sim 10^{36}$ accelerated electrons per second in the transition region above the magnetic network on the whole solar surface. Direct observational evidence for such accelerated electron groups may verify whether the above reconnection processes does or does not heat the corona.

In the solar transition region, these current sheets are only local small-scale structures, and they are discontinuous and dispersed. Gordovskyy et al. (2010) found that protons and electrons can be accelerated to very high energies (up to tens of MeV) in a small transient magnetic reconnection event [38]. It is obvious that the non-thermal particles generated by magnetic reconnection acceleration in the solar transition region are locally and anisotropically distributed.

### 2.2. Radio Emissions by Non-Thermal Electrons in the Transition Region

Since there are indeed non-thermal electrons in the solar transition region, we have to ask, do they emit radio emission? How do these non-thermal electrons generate radio emission?

As we discussed above, the plasmas in the transition region with relatively strong magnetic fields still have β ≪ 1. Additionally, we may calculate that the timescale of collision is much longer than that of gyration. The non-thermal electrons have enough time to emit radio waves before they are thermalized by collision. At the same time, after all, the plasma in the solar transition region is denser than that in the corona, and here, the non-thermal electrons are more easily thermalized by collision. Therefore, the lifetime of radio emission generated in the transition region will be shorter than those in the corona, and we will collectively refer to them as transition region small-scale microwave bursts (TR-SMBs), i.e., the radio signals of non-thermal processes that occur in the transition region. Then, what is the dominant emission mechanism of TR-SMBs?

Firstly, the bremsstrahlung emission is a possible candidate. As we mentioned above when discussing the range of parameters, different from the partial ionized chromospheric plasmas, the transition region comprises almost fully ionized plasma. For example, according to the results of Vernazza et al. (1981), at a height of 2154 km (nearly corresponding to the bottom of the transition region) above the photosphere, T = 20,700 K,

$n_H = 1.205 \times 10^{10}$ cm$^{-3}$, and $n_e = 1.211 \times 10^{10}$ cm$^{-3}$ [23]. In the chromosphere, the $H^-$ process is the major mechanism at low altitude (below 500 km), and the classical bremsstrahlung mechanism becomes the major emission at higher altitudes [39]. However, in the transition region, the temperatures exceed 20,000 K, and neutral particles without ionization can be almost negligible. In this case, the opacity of the bremsstrahlung radio emission can be approximately expressed as [25] k(f) $\approx 0.2 \frac{n_e^2}{f^2 T^{\frac{3}{2}}}$ (cm$^{-1}$). Here, $n_e$ is the electron density (unit of cm$^{-3}$), $f$ is the radio frequency (Hz), and T is the temperature (K). Considering the typical plasma density and temperature in the transition region, we know that it is optically thin at centimeter–millimeter wavelengths.

In the solar transition region, magnetic fields are ubiquitous and stronger than the coronal magnetic field above the flaring source region. As we know the emissivity of bremsstrahlung is $\eta_f \propto \frac{n_e^2}{T^{1/2}}$ while the emissivity of gyroresonance and gyrosynchrotron is $\eta_f \propto n_e T^\alpha B^\beta$, here, $\alpha > 1$, $\beta > 1$, and B is the magnetic field strength [25]. In the transition region above some sunspot regions or magnetic network regions, due to the strong magnetic field and the increasing temperature, the gyroresonance emission, especially the gyrosynchrotron and synchrotron emission by non-thermal electrons can easily surpass the bremsstrahlung emission and become the dominant emission mechanism, and the bremsstrahlung emission can be negligible. Therefore, when discussing the non-thermal signals of the transition region, it is also apt to discuss the gyroresonance emission of non-thermal electrons.

Considering the magnetic field strength in the solar transition region, the gyrofrequency ($f_{ce}$[MHz] $= 2.8B$[G]) ranges from about 300 MHz (in the quiet solar region) to about 8 GHz (around some network regions and sunspot regions). Here, B is the magnetic field strength in the transition region. As for the non-thermal electrons with Lorentz factor ($\gamma$), the typical frequency of gyroresonance emission can be approximately expressed as follows:

$$f[\text{GHz}] \approx 2.8 \times 10^{-3} \gamma^3 B[\text{G}] \tag{2}$$

In the quiet region, we may suppose B $= 100$ G; if the energy (E) of non-thermal electrons is 300 keV, then the emission frequency (f) should be about 1.5 GHz, and when $E_{acc} = 500$ keV, f $\sim 6$ GHz. When $E_{acc} = 1.0$ MeV, f $\sim 20$ GHz. In the transition region above the networks of the quiet Sun, the magnetic field should be 500 G, and the above corresponding emission frequencies should be 7.3 GHz, 30 GHz, and 100 GHz for E $= 300$ keV, 500 keV, and 1.0 MeV, respectively. They all fall in the range of centimeter–millimeter wavelengths. In the transition region above the active sunspot region, we may suppose that B $= 2000$ G. When $E_{acc} = 300$ keV, we may record f $\sim 38$ GHz; when $E_{acc} = 500$ keV, we have f $\sim 123$ GHz, and when $E_{acc} = 1.0$ MeV, f $\sim 388$ GHz. They all fall in the millimeter-wave range. With the above parameter assumptions and the estimation of the number of accelerated electrons, it is further possible to estimate the emission flux intensity in the range of 0.2–2 solar flux unit (sfu) at a frequency of 35 GHz. Compared to the quiet Sun flux intensity of about 2400 sfu, a sensitivity better than 0.01–0.1% is required, and the dynamic range of the telescope should be higher than 30–40 dB.

Due to the thickness of the solar transition region with the above parametric range being only a few hundred km, the accelerated electrons may fly out of the transition region in just millisecond timescales. Therefore, the lifetime of the related millimeter-wave emission is also in millisecond timescales. On the contrary, the flare-accelerated electrons may fly in the corona for tens of seconds or minutes, and the lifetime of the related type III radio bursts will range from seconds to minutes. This requires the temporal resolution of the telescope to be in sub-milliseconds; otherwise, the signal will be averaged, smoothed out, and undetectable.

Overall, the typical frequency range of the radio emission of the non-thermal electrons from the solar transition region should be from several GHz to about hundred GHz, that is to say, from centimeter wave to millimeter wave. Higher frequency millimeter waves may

come from the solar chromosphere below the transition region. For example, White et al. (2017) discussed observations derived from ALMA's 12 m single-dish fast-scan maps of the quiet Sun and derived a chromospheric temperature of 7300 K at 100 GHz and 5900 K at 239 GHz [40]. Nindos et al. (2021) applied the ALMA's imaging observations at 100 GHz and 239 GHz to study the quiet solar region and found many transient brightening events also coming from the chromosphere [41].

Obviously, the spectral characteristics of the above centimeter–millimeter-wavelength emission depends on the spectral energy of the non-thermal electrons when dominated by the acceleration mechanism. As we discussed in Section 2.1, in the transition region, particle accelerations are mainly associated with turbulences or magnetic reconnections. The turbulent acceleration of electrons can occur randomly in the transition region, and the distribution should be isotropic, and their propagation distance is inevitably not very far. When the generated non-thermal electrons propagate in the transition region with a high gradient of magnetic fields, they will generate radio bursts with significant frequency drifting-rates, which are possibly groups of spike-like bursts or narrowband type III-like bursts. Each individual burst will have very narrow frequency bands and very short timescales. Based on the scaling law extrapolation [42] of centimeter–decimeter spike bursts, if there are spike-like bursts in the millimeter wavelengths, the typical frequency bandwidth is about 1% of its center frequency, and the typical lifetime is about 0.5–1.0 ms. The magnetic reconnection accelerating electrons in the solar transition region are related to small-scale current sheets and produce non-thermal electron beams. These non-thermal electrons may generate small-scale centimeter–millimeter-wave bursts, such as groups of narrowband type III-like bursts, similar to the small-scale microwave bursts that occur in decimeter wavelengths [43]. Here, different from the decimeter-wavelength narrowband type III bursts which are generated from the coherent plasma emission mechanism, the transition region small-scale centimeter–millimeter-wavelength narrowband type III-like bursts or spike-like bursts are generated from the interaction between the non-thermal electrons and the strong magnetic fields, and the emissions are incoherent and can be named transition region small-scale microwave bursts (TR-SMBs). Their frequency drifting rates can be expressed follows:

$$\frac{df}{dt} = 2.8 \times 10^{-3} \gamma^3 \frac{dB}{dr} v_b \qquad (3)$$

Here, the unit of frequency drifting rate ($\frac{df}{dt}$) is GHz·s$^{-1}$. $\frac{dB}{dr}$ is the magnetic gradient with G·km$^{-1}$, and $v_b$ is the velocity of the non-thermal electrons with km·s$^{-1}$. When E = 300 keV, $v_b = 0.82\ c$; when E = 500 keV, $v_b = 0.91\ c$, and so on. Approximately, we may simply assume $v_b \approx c$; here, $c$ is the light speed. This indicates that these non-thermal electron beams may escape from the solar transition region in a timescale of 0.1–0.5 ms. The lifetime of the related radio bursts should match this timescale.

It is obvious that the frequency drifting rate is not only proportional to the magnetic field gradient in the source region but also related to the energy of non-thermal electrons. As the energy increases, the frequency drift rate increases rapidly. Supposing the magnetic gradient in the transition region is $\frac{dB}{dr} \approx 0.01$ G·km$^{-1}$ [44], then $\frac{df}{dt} \approx 45$ GHz·s$^{-1}$ for E = 300 keV, and 185 GHz·s$^{-1}$ for E = 500 keV, and so on. These estimations show that TR-SMBs should have much more rapid frequency drifting rates than that of type III bursts at centimeter–decimeter and even meter wavelengths. This also requires that the temporal resolution (Δt) and frequency resolution (Δf) of a proposed broadband millimeter-wave dynamic spectrometer have the ability to identify such high-frequency drift rates. For example, when Δt = 1 ms, then Δf < 200 MHz.

Obviously, the signals of RS-SMBs are the components of the background quiet Sun flux intensity in the centimeter–millimeter wavelengths, equivalent to the fluctuations in the background of the thermal emission. Due to the fact that the thickness of the solar transition region is only a few hundred km, the source region of TR-SMBs must be much

smaller, meaning that the existing telescopes cannot directly identify them through imaging observations, and it is unlikely that even the telescopes developed in the future will be able to identify them. Telescopes with insufficient temporal and frequency resolutions will average the records and smooth out the non-thermal signals. A suitable approach for detecting TR-SMBs is to use broadband dynamic spectrum observations with a broadband frequency range of 20–100 GHz, high temporal resolution $\Delta t < 0.2$–0.5 ms, high frequency resolution $\Delta f < 50$–300 MHz, high sensitivity < 0.2 sfu, and large dynamic range > 30 dB. Imaging observations are not necessary. Figure 2 presents the possible pattern of the radio spectrogram of the transition region. From observing the centimeter–millimeter-wave broadband dynamic spectrum of the quiet Sun; identifying TR-SMBs, extracting their bandwidth, lifetime, frequency drifting rate, and other related parameters; and studying their distribution and evolution patterns, it is possible to reveal the non-thermal energy release mechanism in the solar transition region. This result can help us to clarify the sources of mass and energy required for coronal heating and solar eruptions.

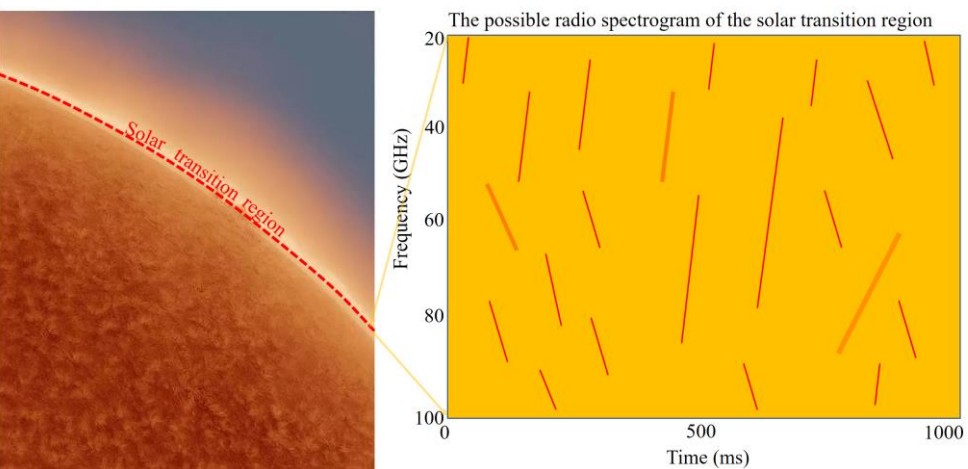

**Figure 2.** The possible radio spectrogram of the solar transition region. Here, the red lines present the possible TR-SMBs overlapping in the background emission of the quiet Sun (yellow).

## 3. Limitations of Radio Observations of the Transition Region

Given that observations from broadband centimeter–millimeter-wave dynamic spectrometers are so important for revealing the physical nature of the non-thermal processes in the solar transition region, what is the current international observation status?

Unfortunately, there are currently no broadband dynamic spectrometers specifically designed for observing the solar transition region available in the world. This major limitation is because of the strong absorption of water and oxygen molecules in the Earth's atmosphere in the centimeter–millimeter wavelength, making observations in our proposed frequency range impossible. Figure 3 presents the averaged atmospheric absorption index of the radio emissions at different frequencies. Here, we can see several strong absorption peaks. Overall, the peaks at 50–75 GHz and 105–125 GHz are the strong absorption peaks derived from oxygen molecules, and the peaks at 175–220 GHz and 300–385 GHz are the strong absorption peaks derived from water molecules. Additionally, the peak at 20–26 GHz is a relatively weak but still obvious absorption peak derived from water molecules. It is precisely because of the existence of these absorption peaks that we are unable to construct a ground-based telescope system to obtain observations of the solar broadband dynamic spectrum in the millimeter wavelength. Although there are many ground-based solar millimeter-wave telescopes in the world (Table 1), they are only able to facilitate observations at a few narrowband frequencies (Table 1).

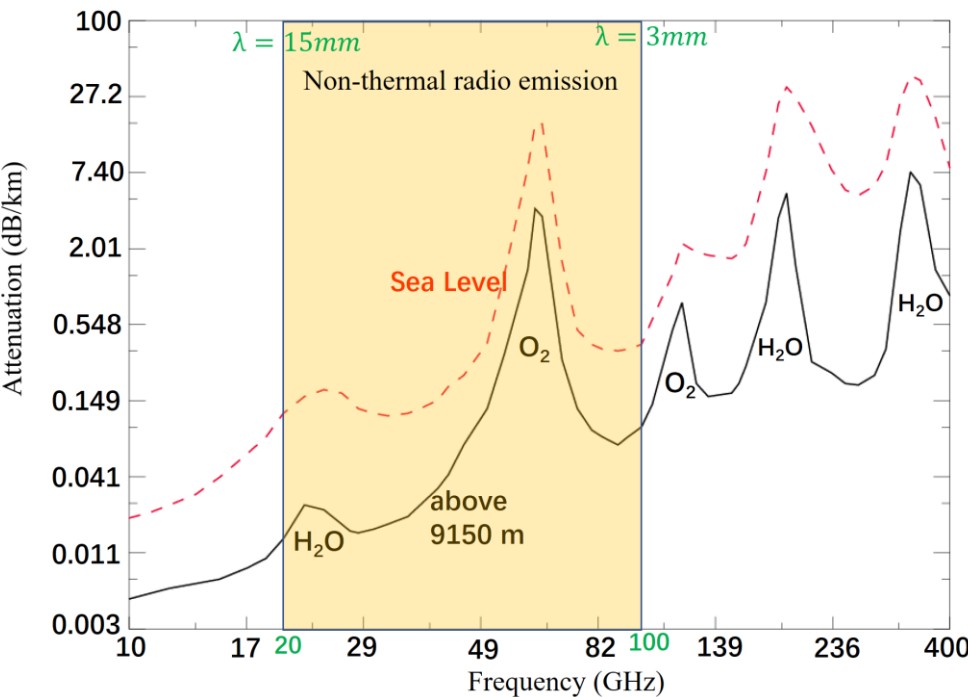

**Figure 3.** The averaged atmospheric absorption index in millimeter wavelengths at sea level and at a height of 9150 m above the sea. The yellow part shows the frequency range of the proposed Solar Ultra-wide Broadband Millimeter-wave Spectrometer (SUBMS). The green characters represent the upper and lower limits of the frequency range and the corresponding wavelengths of the SUBMS.

**Table 1.** List of the existing solar millimeter-wave telescopes.

| Telescope | Frequency (GHz) | Status |
|---|---|---|
| Nobeyama RadioHiograph (NoRH), Japan [45] | Imaging at 17, 34 | Closed in 2020 |
| Nobeyama Radio Polarimeters (NoRP), Japan [46] | 1.0, 2.0, 3.75, 9.4, 17, 35, 80 | Daily (80 GHz; has been closed since 2012). |
| Itapetiga radome-enclosed telescope, Brazil [47] | 7.0, 23.2, 30, 44.3, 90 | Occasional observation based on events |
| Solar Submillimeter Telescope (SST), Argentina [48] | 212, 405 | Occasional observation based on events |
| Berkeley-Illinois-Maryland Array (BIMA), USA [49] | Imaging at 86 | Occasional observation based on events |
| Atacama Large Millimeter/submillimeter Array, ALMA, Czeli [50–54] | Imaging at 100, 239 | Solar observation time: 2–3% |
| Chashan Broadband Solar millimeter spectrometer (CBS), China [55] | 35–40 with frequency resolution: 0.5 GHz | Daily observations since 2022 |

Table 1 shows that there are no solar-dedicated broadband millimeter-wave dynamic spectrometers in the world which cover the whole frequency range of TR-SMBs. The Chashan Broadband Solar millimeter spectrometer (CBS) from China [55,56] was the first solar broadband millimeter-wave dynamic spectrometer. It has a temporal resolution of 134 ms and a frequency resolution of 153 kHz, which may be integrated into 500 MHz. It can obtain the clear profiles and spectra of solar flares [56]. However, its frequency bandwidth is only 5 GHz (35–40 GHz), which is not wide enough to cover the wavelengths of TR-SMBs, and the temporal resolution (134 ms) is not short enough to identify TR-SMBs.

Therefore, the CBS can be used to observe solar flares, but it is not suitable to observe the signals of non-thermal processes in the transition region. In fact, we need a much wider broadband centimeter–millimeter-wave dynamic spectrometer that can cover a frequency range from about 20 GHz to 100 GHz with a bandwidth of 80 GHz.

## 4. The Proposed Solar Ultra-Wide Broadband Millimeter-Wave Spectrometer

Here, we propose a new type of Solar Ultra-wide Broadband Millimeter-wave Spectrometer (SUBMS) that can be used to observe TR-SMBs. SUBMSs should cover the typical frequency range associated with the non-thermal electrons produced in the solar transition region (from 20 GHz to 100 GHz, see the yellow part in Figure 3) with high temporal resolution, high frequency resolution, and high sensitivity but without the capability to facilitate imaging. The beam size of the antenna of our proposed spectrometer just covers the full solar disk (32 arc-minute), and the pointing accuracy is designed to be <2 arc-minute. The main parameters of the SUBMS are listed in Table 2.

**Table 2.** The main parameters of the proposed space-based Solar Ultra-wide Broadband Millimeter-wave Spectrometer.

| | |
|---|---|
| Observing frequency | 20–100 GHz (4 elements: 20–40, 40–60, 60–80, 80–100 GHz) |
| radius of antenna | 98.3 cm, 65.5 cm, 49.2 cm, 39.3 cm, respectively |
| Antenna design | Front-fed parabolic reflector antenna, Cassegrain dual-reflector antenna |
| Feed design | Quadruple-ridged flared horn or Corrugated Horn |
| Dynamic range | >30 dB |
| Polarization | RCP and LCP |
| Sensitivity | <0.2 sfu |
| Frequency resolution | 100 MHz, 200 MHz, 250 MHz, 400 MHz, respectively |
| Frequency channel | 200, 100, 80, 50, respectively. A total of 430 channels |
| Time resolution | 0.2 ms |
| Calibration | Using the Y Factor method with $T_{hot}$ from noise input and $T_{cold}$ from cold sky |

As we discussed in Section 3, because of the strong absorption of oxygen and water molecules in the frequency range of 20–100 GHz, the proposed SUBMS cannot work on the ground. Therefore, we have to put it on some spacecrafts, namely the China Space Station, which has an orbital altitude of 400–450 km and an inclination angle of 42–43°. On such an orbit, the telescope can completely avoid the absorption of water and oxygen molecules and the transparency fluctuations of the Earth's atmosphere, thus achieving complete and stable observations of the Sun in full frequencies. The receiver adopts a certain temperature controlling method to minimize the impact of temperature changes on observations as much as possible. Table 2 lists the main designed parameters of a space-based SUBMS. We plan to select the front-fed parabolic reflector antenna and quadruple-ridged flared horn as the feeds for frequencies of 20–40 GHz and 40–60 GHz, while we plan to select the Cassegrain dual-reflector antenna and Corrugated Horn as the feeds for frequencies of 60–80 GHz and 80–100 GHz.

Practically, a space telescope costs a lot. For safety, before the space-based SUBMS can be launched into the space, it is necessary to select some frequency bands for observational experiments on the ground in certain high-altitude and dry sites. From Figure 2, we can find that 20–50 GHz and 75–100 GHz are relatively suitable windows for making observations using ground-based telescopes. Because CBS already covers the frequency range of 35–40 GHz, we will select 20–35 GHz as a suitable band for constructing the ground-based test broadband millimeter-wave spectrometer (test-SUBMS). As for the site, we may select

Saisteng Mountain in Qinghai province, China. Here, there is a ready-made astronomical observation station, and the altitude is 4030 m. This location is far from big cities and the modern industrial centers, and the air in this location is very clean and very dry, with annual humidity being < 10%. The environment temperature is relatively stable, which may ensure the gain stability of the receiving system. As we have numerous millisecond-lifetime observational targets and the fluctuation in the environmental temperature is always at a timescale of a minute or even longer, we can ignore the atmospheric transparency fluctuations. However, we still plan to install a temperature-controlling system in the receiving system to minimize the impact of environmental temperature changes on observations. Additionally, we will also consider the corrections of atmospheric absorption by using an atmospheric absorption model [57–59]. Table 3 lists the main parameters of the test-SUBMS.

**Table 3.** Main parameters of the test ground-based broadband millimeter-wave spectrometer (test-SUBMS).

| | |
|---|---|
| Observing frequency | 20–35 GHz, or 75–100 GHz |
| radius of antenna | 112.4 cm, or 39.3 cm |
| Antenna design | Front-fed parabolic reflector antenna, Cassegrain dual-reflector antenna |
| Feed design | Quadruple-ridged flared horn or Corrugated Horn |
| Dynamic range | >30 dB |
| Polarization | RCP and LCP |
| Sensitivity | <0.2 sfu |
| Frequency resolution | 50 MHz or 250 MHz |
| Frequency channel | 300 or 100 |
| Time resolution | 0.2 ms or 0.4 ms |
| Construct site | Saisteng Mountain in Qinghai, China (altitude: 4030 m) |
| Calibration | Using the Y factor method with $T_{hot}$ from noise input and $T_{cold}$ from cold sky |
| Atmospheric Absorption correction | Using an atmospheric absorption model [58,59] |

The calibration of both the space-based SUBMS and the ground-based test-SUMBS will be carried out by using the relative calibrating method [57], and the reference source will be the cold sky and a known constant noise source.

We will complete our experiment involving the test-SUBMS in the autumn of 2024. Then, the Sun will come into its peak year of solar cycle 25, and the test-SUBMS will also be able to observe the flare-associated millimeter-wave bursts of the high-energy processes in the corona, as well as the centimeter–millimeter-wave emissions of the non-thermal energy release in the transition region without flares. As the TR-SMBs are possibly very weak, in the observation experiment, we will attempt to adjust the observation parameters of the telescope, including increasing the number of antenna elements to improve the sensitivity of the observations. If, after the ground-based experiment, we have successfully obtained observation data for the broadband millimeter-wave spectrum of the transition region using the test-SUBMS, then we may consider using the space-based SUBMS at the full frequency range.

## 5. Conclusions

The solar transition region is the key to revealing the mysteries of coronal heating, the nature of solar eruptions, and the origin of solar wind. Here, the huge gradient of temperature and density inevitably drives strong convective motion, which, in turn, drives the complex movement of magnetic field lines and generates various non-thermal energy releases, including non-thermal electron beams which may produce groups of spike-like

bursts or narrowband type III-like bursts in the centimeter and millimeter wavelengths (TR-SMBs). Unlike radio bursts during solar flares, TR-SMBs are large in number; have weak signals, short lifetimes (in millisecond timescale), and rapid frequency drift rates; and cover a frequency range of about 20–100 GHz. They are the typical signals of the non-thermal energy release in the transition regions. Carrying out observations using broadband centimeter–millimeter-wave dynamic spectrometers with high temporal resolution and high frequency resolution represents a unique approach to obtaining the dynamic information of TR-SMBs.

This work proposed a new Solar Ultra-wide Broadband Millimeter-wave Spectrometer (SUBMS) for observing TR-SMBs. Due to the absorption of water and oxygen molecules in the Earth's atmosphere, it is very hard to obtain the broadband millimeter-wave dynamic spectrum in the full frequency range from ground-based telescopes. We have to launch the proposed telescope into space beyond the Earth's atmosphere. However, with careful site selection, broadband dynamic spectrum observations can also be obtained via using ground-based telescopes in some frequency bands. Prior to launching the space-based SUBMS, we will build an experimental ground-based system (test-SUBMS) in a frequency range of 20–35 GHz or 75–100 GHz in Saisteng Mountain in Qinghai province, China. Actually, such an instrument can observe TR-SMBs under quiet solar conditions, as well as the eruptive high-energy processes during solar flares. Once some valuable broadband dynamic spectrum observations have been obtained from the above-mentioned test-SUBMS, then we will have enough confidence to develop and launch the space-based SUBMS to observe TR-SMBs in a full frequency range. Although these telescopes have no imaging capabilities, we can still identify TR-SMBs in broadband dynamic spectrum and analyze their distribution and variation characteristics. At that time, we will be able to truly reveal the energy release mechanism and mass propagation and transfer process in the solar transition region and subsequently provide crucial evidence for explaining coronal heating and elucidate the origins of solar eruptions and solar winds.

**Author Contributions:** Conceptualization, B.T. and J.H.; methodology, F.L., J.F., L.C. and J.S.; formal analysis, B.T., J.H. and Y.Z.; writing—original draft preparation, B.T.; writing—review and editing, B.T.; supervision, Y.D.; project administration, Y.D.; funding acquisition, B.T., Y.D. and L.C. All authors have read and agreed to the published version of the manuscript.

**Funding:** This research was funded by the National Key R&D Program of China (Nos. 2021YFA1600503, 2022YFF0503001, 2022YFF0503800), a Program of the National Science Foundation of China (Nos. 11973057, 12173050), and the Strategic Priority Research Program of the Chinese Academy of Sciences, Grant No. XDB0560302.

**Data Availability Statement:** Data are contained within the article.

**Acknowledgments:** This work was also supported by the international collaboration between ISSI-BJ and the International Partnership Program of the Chinese Academy of Sciences (grant number 183311KYSB20200003).

**Conflicts of Interest:** The authors declare no conflict of interest. The funders had no role in the design of the study; in the collection, analyses, or interpretation of data; in the writing of the manuscript; or in the decision to publish the results.

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
