# Peer review of "The Non-Thermal Radio Emissions of the Solar Transition Region and the Proposal of an Observational Regime"

_universe, doi:10.3390/universe10020082_

Round 1

Reviewer 1 Report

Comments and Suggestions for Authors

The article, 'Radio Emission and Observations from the Solar Transition Region,' provides a thorough examination of the challenges inherent in millimeter wavelength observations within the solar transition region. It systematically addresses the potential mechanisms responsible for radio emissions in this solar domain. Additionally, the article introduces a new instrument concept, offering a prospect for advancements in solar observations. The scientific discourse underscores the intricacies of millimeter wavelength observation, elucidates the underlying processes of radio emission, and presents a forward-looking approach through the proposed instrument design.

Here are some comments:

- In the introduction the author state that transition region is the most peculiar layer. What makes it peculiar? 

- In line 38, change "the cold, partially" by "the colder, partially"

- line 46, the author talk about coronal heating. It will be good to remind the readers the different existing coronal heating mechanisms in the literature.

- line 54, what are these "transient activities" in the transition region?

- lines 65-68. The radio emission is an important however is not the most important. There is no such things as the most important emission. Here it will be include a small discussion on opacity and optical depth. Also, will be good to start a discussion on the different mechanism of radio emission and the dominance depending on the local conditions (density, magnetic field strength, etc.).

- lines 70-72, although in general the idea is true, the reason is not related to the height, is the dominance of the radio emission generation mechanism over the others. See Chen papers.

- line 86, density was not discussed in section 1.

- lines 103-104, How different is the relation to the one observed between HXR and GHz?

- lines 111-112 Are you suggesting here that the emission from flares at mm-wavelengths is produced in the corona, near the reconnection region?

-section 2.2 I find the whole discussion disconnected of the text. What is the objective of this section?

lines 204-208, what is the influence of the assumption made in the drift rates? Do you see and appreaciable difference?

Table 1. It is not Japen, but Japan.

Section 4

I think the whole section requires a better description of the instrument concept. Is it a radio polarimeters? Is it an interferometer? Does it have imaging capabilities? The units in space, what is the optimal orbit? what are the pointing requirements? How sensitive to temperature variations is the measurements? If so, how you will account for that?

Comments on the Quality of English Language

I think the article in general is well written, but it will benefit from a revision.

Author Response

Reply to the Review report 1:

The article, 'Radio Emission and Observations from the Solar Transition Region,' provides a thorough examination of the challenges inherent in millimeter wavelength observations within the solar transition region. It systematically addresses the potential mechanisms responsible for radio emissions in this solar domain. Additionally, the article introduces a new instrument concept, offering a prospect for advancements in solar observations. The scientific discourse underscores the intricacies of millimeter wavelength observation, elucidates the underlying processes of radio emission, and presents a forward-looking approach through the proposed instrument design.

Reply: Firstly, the authors greatly appreciate the reviewer's affirmation and helpful suggestions for this article. We have carefully and meticulously revised the entire text based on the comments of the reviewers, rewriting most of the paragraphs and correcting previous inaccurate descriptions. Now, the entire text should appear more reasonable and complete. In the following, we will answer the questions raised by the reviewer one by one.

Here are some comments:

- In the introduction the author state that transition region is the most peculiar layer. What makes it peculiar? 

Reply: The solar transition region becomes unique due to its highest gradients of physical parameters. Its formation should be related not only to the process of mass and energy input from the Sun's interior, but also to the heat conduction downwards from the upper hot corona. However, it is still not very clear how the transition region is formed, and we need to conduct deeply research on the energy releasing mechanism and transportation in this region.

- In line 38, change "the cold, partially" by "the colder, partially"

Reply: OK. We have corrected this spelling error In the revised version.

- line 46, the author talk about coronal heating. It will be good to remind the readers the different existing coronal heating mechanisms in the literature.

Reply: OK. In the revised version, we added 8 references and the following sentences to mention the existing coronal heating mechanisms: “We know that the mystery of coronal heating is one of the eight major challenges in contemporary astronomy [4, 5]. The existing heating mechanisms can be roughly divided into three categories: wave heating [6, 7], reconnection heating [8, 9], and magnetic field gradient pumping heating [10, 11]. However, there is currently no consensus on which mechanism truly dominates the coronal heating process.”

- line 54, what are these "transient activities" in the transition region?

Reply: Sorry, the expression in our original text was not very accurate. In the revised version, we rewrite this paragraph as follows: “Observations show that there are many small-scale transient activities which may affect the energy releasing and mass transfer in the solar transition region, such as spicules, small-scale jets, various small bright points, and even nanoflares.”

- lines 65-68. The radio emission is an important however is not the most important. There is no such things as the most important emission. Here it will be include a small discussion on opacity and optical depth. Also, will be good to start a discussion on the different mechanism of radio emission and the dominance depending on the local conditions (density, magnetic field strength, etc.).

Reply: In the revised version, we rewrite this paragraph as follows: “All non-thermal processes are associated with the generation and propagation of energetic particles, which interact with plasmas and magnetic fields to produce radio emission. Therefore, radio observation also will play an extremely important role for detecting non-thermal processes in the solar transition region. As we will discuss in Section 2, the non-thermal electrons will generate millimeter-wave emission in the transition region mainly by gyrosynchrotron mechanism and partly by bremsstrahlung mechanism. Considering the magnetic field, plasma density and temperature, the emission is optically thin and the spectrum is directly related to the energy distribution of the electrons.”

- lines 70-72, although in general the idea is true, the reason is not related to the height, is the dominance of the radio emission generation mechanism over the others. See Chen papers.

Reply: In the revised version, we rewrite this part into: “Besides the dominance of the emission mechanism and propagation effect [19], generally, the shorter the wavelength, the closer the source region of radio emission is to the solar surface; the longer the wavelength, the farther away the source region of radio emission from the solar surface.” Here, we also added the reference of Chen et al. (2023).

- line 86, density was not discussed in section 1.

Reply: we rewrite this sentence into: “In the solar transition region, the plasma density rapidly decreases from about   to  ”.

- lines 103-104, How different is the relation to the one observed between HXR and GHz?

Reply: Yes, there are obvious difference between radio bursts at different frequencies. According to the work of Kaufmann et al. (1985), the HXR bursts are almost identical to the radio bursts at 90 GHz, similar to the radio bursts at 30GHz, but nearly no corresponding signals at 7 GHz. Here, we rewrite this part into: “Accompanying with these millimeter-wave bursts, almost identical hard X-ray temporal structures are simultaneously observed. For example, Kaufmann et al. reported that intense and fast pulsed structure at 90 GHz and relatively smaller amplitude structure at 30 GHz were almost identical to the temporal structures of hard X-ray bursts at energy of 24-108 keV at 13:26 UT on 1984 May 21. At the same time, there was almost no enhancement at 7 GHz corresponding to the hard X-ray bursts during the same period [25].”

- lines 111-112 Are you suggesting here that the emission from flares at mm-wavelengths is produced in the corona, near the reconnection region?

Reply: In fact, most flare source regions are located in the corona, but there are also some flare source regions that may be located in lower transition regions or even in the chromospheres. Therefore, we rewrite this sentence as: “We know that compared to the most flaring source region mainly locating in the corona”.

-section 2.2 I find the whole discussion disconnected of the text. What is the objective of this section?

Reply: In fact, this part is closely related to the text. Because here, we discuss the possible generation mechanisms of non-thermal electrons in the solar transition region, as well as the emission mechanisms and spectral characteristics of these non-thermal electrons. This is the basis for our discussion on how to observe and select the parameters of proposed telescope – the Solar Ultra-wide Broadband Millimeter-wave Spectrometers. We have rewritten the content of this section and added many related content to illustrate the above purpose.

lines 204-208, what is the influence of the assumption made in the drift rates? Do you see and appreciable difference?

Reply: Equation (2) indicates that the frequency drifting rate is not only proportional to the magnetic field gradient in the source region, but also related to the energy of non-thermal electrons. As the energy increases, the frequency drift rate increases rapidly. Here, we just assume several cases to estimate the possible magnitude of frequency drifting rates of transition region small-scale millimeter-wave bursts (TR-SMBs), which may provide a basic clue for selecting the frequency resolution and time resolution of the proposed Solar Ultra-wide Broadband Millimeter-wave Spectrometers.

Table 1. It is not Japen, but Japan.

Reply: We correct it in the revised version.

Section 4

I think the whole section requires a better description of the instrument concept. Is it a radio polarimeters? Is it an interferometer? Does it have imaging capabilities? The units in space, what is the optimal orbit? what are the pointing requirements? How sensitive to temperature variations is the measurements? If so, how you will account for that?

Reply: Due to the fact that the thickness of the whole solar transition region is only a few hundred km, the source region of TR-SMBs must be much smaller, so that our existing and even future telescopes cannot directly identify them through imaging observations. The most effective approach to detect the TR-SMBs is through broadband dynamic spectrum observations with high temporal resolution, high frequency resolution, and high sensitivity and dynamic range. There is no need for imaging observations. By observing the millimeter-wave broadband dynamic spectrum of the quiet Sun, identifying TR-SMBs and studying their distribution and evolution patterns, it is possible to reveal the non-thermal energy release mechanism in the solar transition region, which may help us to clarify the sources of mass and energy required for coronal heating. Therefore, our proposed Solar Ultra-wide Broadband Millimeter-wave Spectrometers (SUBMS) is only a broadband spectrometer, without imaging capabilities. The beam size of the antenna is just covering the full solar disk (32 arc-minute) and the pointing accuracy of the telescope is < 2 arcminute. The receiver will adopt certain temperature controlling method to minimize the impact of temperature changes on observations as much as possible. The optimal space platform is the China Space Station, with an orbital altitude of 400-450 km and an inclination angle of 42-43°. On such an orbit, the detector can completely avoid the absorption of water and oxygen molecules in the Earth's atmosphere, thus achieving complete and stable observation of the sun in all frequencies.

Reviewer 2 Report

Comments and Suggestions for Authors

The paper is interesting and presents what one would expect in terms of high frequency radio emission from the solar transition region. The paper content is then used to propose in future a new Solar Ultra-wide Broadband Millimeter-wave Spectrometers (SUBMS) in space. As a first step, the authors propose the construction of an experimental ground-based broadband millimeter-wave spectrometers in the frequency range of 20-35 GHz or 75-100 GHz.

Again, I believe the paper focus on important solar physics questions and deserves publication. But not in the present form. I list below some major and minor comments which need to be taken into account, for the sake and clarity of the manuscript. 

1) Important Missing references: I found the following important references on high-frequency radio and IR solar observations, case events and/or simulations, characteristics of flare and non-flare processes in the lower layers of the solar atmosphere, the authors need to be aware of.  

- White, S.M. et al. 2017, Solar Physics, Observing the Sun with the Atacama Large Millimeter-submillimeter Array (ALMA): Fast-Scan Single-Dish Mapping

to include in the 2nd paragraph of section 2 (lines 86-96)

- Warren, H.P. et al., 2016, ApJ, Transition Region and Chromospheric Signatures of Impulsive Heating Events. I. Observations

- Reep, J.W. et al., 2016, ApJ, Transition Region and Chromospheric Signatures of Impulsive Heating Events. II. Modeling

to include in the 2nd paragraph of the Introduction (lines 54-68)

2) In section 2.1, authors mention the acceleration of particles by strong turbulence. This is a known possibility in the lower and higher coronal plasma where BETA plasma is << 1. Are the authors believe that this process is still efficient in a higher BETA plasma like expected in the transition region ?  are the few hundreds of keV announced in line 135 still valid ? Please comment and explain.

3) also in section 2.1, authors refer to magnetic reconnection for accelerating particles up to few (or few tens) MeV. But what is the electron number accelerated in such an event ? probably to small to detect any radio emission. Please comment and explain.

4) Assuming you have the energetic (accelerated) electrons, how sure are you that they won't be thermalized by collisions in the (dense) transition region, before radiation any radio waves ? Please comment and explain.

5) More generally, in the higher chromosphere/transition region we expect a strong thermal (free-free) emission in the radio wavelength range (few GHz up to few 100s GHz). On the other hand, the non-thermal emission the authors plan to observe are in general very compact sources, representing a very faint excess (above quiet Sun). How do authors plan to detect such small excesses ? I understand from Tables 2 and 3 that, given the diameters of the antennae, HPBW will be large, covering areas >> active region size. Will the emission from bright compact sources be detectable against the thermal background ?

6) other comments:

- in Table 1, for the SST instrument, use the following reference (rather than 25)

 Kaufmann, P. et al., 2008, IEEE, New telescopes for ground-based solar observations at sub-millimeter and mid-infrared

This is the official reference for the SST (Solar Submillimeter Telescope) in Table 1

Throughout all text please do not use "Type III bursts" for the expected millimeter (incoherent) emission from electron beams. I understand that authors are NOT confused with the well-known (coherent) Type IIIs at dm-m wavelengths. But even so, please use another wording, e.g. millimeter emission from electron beams.

Comments on the Quality of English Language

Only a small editing of the English language is needed.

Author Response

Reply to the Review report 2

The paper is interesting and presents what one would expect in terms of high frequency radio emission from the solar transition region. The paper content is then used to propose in future a new Solar Ultra-wide Broadband Millimeter-wave Spectrometers (SUBMS) in space. As a first step, the authors propose the construction of an experimental ground-based broadband millimeter-wave spectrometers in the frequency range of 20-35 GHz or 75-100 GHz.

Again, I believe the paper focus on important solar physics questions and deserves publication. But not in the present form. I list below some major and minor comments which need to be taken into account, for the sake and clarity of the manuscript.

Reply: Firstly, the authors greatly appreciate the reviewer's affirmation and helpful suggestions for this article. We have carefully and meticulously revised the entire text based on the comments of the reviewers, rewriting most of the paragraphs and correcting previous inaccurate descriptions. Now, the entire text should appear more reasonable and complete. In the following, we will answer the questions raised by the reviewer one by one.

1) Important Missing references: I found the following important references on high-frequency radio and IR solar observations, case events and/or simulations, characteristics of flare and non-flare processes in the lower layers of the solar atmosphere, the authors need to be aware of.  - White, S.M. et al. 2017, Solar Physics, Observing the Sun with the Atacama Large Millimeter-submillimeter Array (ALMA): Fast-Scan Single-Dish Mapping - to include in the 2nd paragraph of section 2 (lines 86-96).

Reply: In the revised version, we insert this part into Section 2.2 as: “Higher frequency millimeter waves may come from the lower part of the solar chromosphere. For example, White et al. (2017) discussed the observation of ALMA’s 12 m single-dish fast-scan maps of the quiet Sun and derived a chromospheric temperature of 7300 K at 100 GHz and 5900 K at 239 GHz [34]. Nindos et al. (2021) applied the ALMA’s imaging observations at 100 GHz and 239 GHz to study the solar quiet region and found many transient brightening events also coming from the chromosphere [35].”

- Warren, H.P. et al., 2016, ApJ, Transition Region and Chromospheric Signatures of Impulsive Heating Events. I. Observations - Reep, J.W. et al., 2016, ApJ, Transition Region and Chromospheric Signatures of Impulsive Heating Events. II. Modeling --- to include in the 2nd paragraph of the Introduction (lines 54-68)

Reply: In the revised version, we added these two references in Section 1.

2) In section 2.1, authors mention the acceleration of particles by strong turbulence. This is a known possibility in the lower and higher coronal plasma where BETA plasma is << 1. Are the authors believe that this process is still efficient in a higher BETA plasma like expected in the transition region? are the few hundreds of keV announced in line 135 still valid? Please comment and explain.

Reply: Actually, in transition region, the plasma beta is still much smaller than 1. Let’s consider the parameters as following: plasma density decreases from about   to  , temperature increases from about K to around  K, while magnetic field strength ranges from 1000 G to 100 G, then we may calculate the plasma beta  in any cases. Considering the highest gradients of density, temperature and magnetic fields and the possible strong convections, we believe that the particle accelerations by strong turbulence are efficient and may generate non-thermal electrons. As for whether it is possible to generate hundreds of KeV high-energy electrons, that is our speculation that needs to be verified through new observations.

3) also in section 2.1, authors refer to magnetic reconnection for accelerating particles up to few (or few tens) MeV. But what is the electron number accelerated in such an event? probably to small to detect any radio emission. Please comment and explain.

Reply: Here, we cited the work of Gordovskyy et al. (2010) where applied the several two-dimensional numerical models of forced magnetic reconnection in the initially force-free Harris current sheet to investigate the particle acceleration and found that protons and electrons can be accelerated to very high energies up to tens of MeV.

4) Assuming you have the energetic (accelerated) electrons, how sure are you that they won't be thermalized by collisions in the (dense) transition region, before radiation any radio waves? Please comment and explain.

Reply: As we discussed above, the solar transition region still has beta  in any cases. Additionally, we may calculate the timescale of collision and gyration and will find that timescale of collision is much longer than that of gyration. Therefore, we are sure that the non-thermal electrons can emit radio waves before they are thermalized by collision. At the same time, after all, the plasma in the solar transition region is denser than that in the corona, and here the non-thermal electrons are more easy to be thermalized by collision. Therefore, the lifetime of radio emission generated in transition region will be shorter than those in the corona, and we will collectively refer to them as transition region small-scale millimeter-wave bursts (TR-SMBs).

5) More generally, in the higher chromosphere/transition region we expect a strong thermal (free-free) emission in the radio wavelength range (few GHz up to few 100s GHz). On the other hand, the non-thermal emission the authors plan to observe are in general very compact sources, representing a very faint excess (above quiet Sun). How do authors plan to detect such small excesses? I understand from Tables 2 and 3 that, given the diameters of the antennae, HPBW will be large, covering areas >> active region size. Will the emission from bright compact sources be detectable against the thermal background?

Reply: In this work, we propose a new telescope of solar ultra-wide broadband millimeter-wave spectrometer (SUBMS) which may cover the full solar disk and the typical frequency range associated with the non-thermal electrons from the solar transition region (Table 2 and 3). It is a broadband dynamic spectrometer with high temporal resolution and high frequency resolution, but without imaging capability. The beam size of the antenna is just covering the full solar disk (32 arc-minute). During the flare time, it can observe the flare-associated millimeter-wave bursts of high-energy processes in the corona. During non-flare time, it will observe the TR-SMBs. As TR-SMBs are short-lived, narrow bandwidth with fast frequency-drifting rates, and even may be very weak. In the observation experiment, we will attempt to adjust the observation parameters of the telescope, including increasing the number of antenna elements to improve the sensitivity of the observation and discover these small-scale energy release events.

6) other comments:

- in Table 1, for the SST instrument, use the following reference (rather than 25) -- Kaufmann, P. et al., 2008, IEEE, New telescopes for ground-based solar observations at sub-millimeter and mid-infrared--This is the official reference for the SST (Solar Submillimeter Telescope) in Table 1

Reply: we correct it in the revised version.

- Throughout all text please do not use "Type III bursts" for the expected millimeter (incoherent) emission from electron beams. I understand that authors are NOT confused with the well-known (coherent) Type IIIs at dm-m wavelengths. But even so, please use another wording, e.g. millimeter emission from electron beams.

Reply: In the revised version, we collectively refer to them as transition region small-scale millimeter-wave bursts (TR-SMBs)

Reviewer 3 Report

Comments and Suggestions for Authors

The core of this manuscript is the suggestion of a solar spectrometer in the 20-100 GHz band, in which the only similar instrument is the Chashan Broadband Solar millimeter spectrometer (CBS),operating in the narrow frequency range of 35-40 GHz. Before that, the authors discuss the chromosphere-corona transition region (Section 1) and its radio emission (Section 2) as well as the current state of instruments capable of observing in the relevant frequency range (Section 3).

I have two problems with this manuscript:

a) Not enough information about the suggested instrument is provided in the text. Antenna design and feeds, type of receivers, absolute calibration, correction for atmospheric absorption variations and possibly others should be added.

b) As a full-disk instrument will only be capable of detecting only flare-associated emission, most of the discussion in Sections 1-2 is not very relevant.

I suggest that the authors completely re-organize their text. If they wish to build their text around the spectrometer proposal they should follow the above suggestions and find a more appropriate title. If, on the other hand, they wish to write about the radio emission from the transition region, they should consider a more extended review article with many more details and references.

Concerning ALMA, the authors may wish to consult the articles https://doi.org/10.3389/fspas.2022.977368, https://doi.org/10.3389/fspas.2022.966444, https://doi.org/10.3389/fspas.2022.981205 and https://doi.org/10.1051/0004-6361/202244532.

Comments on the Quality of English Language

I would prefer to make specific suggestions in the revised manuscript.

Author Response

Reply to the review report 3

The core of this manuscript is the suggestion of a solar spectrometer in the 20-100 GHz band, in which the only similar instrument is the Chashan Broadband Solar millimeter spectrometer (CBS), operating in the narrow frequency range of 35-40 GHz. Before that, the authors discuss the chromosphere-corona transition region (Section 1) and its radio emission (Section 2) as well as the current state of instruments capable of observing in the relevant frequency range (Section 3).

Reply: Firstly, the authors greatly appreciate the reviewer's affirmation and helpful suggestions for this article. We have carefully and meticulously revised the entire text based on the comments of the reviewers, rewriting most of the paragraphs and correcting previous inaccurate descriptions. Now, the entire text should appear more reasonable and complete. In the following, we will answer the questions raised by the reviewer one by one.

I have two problems with this manuscript:

  1. Not enough information about the suggested instrument is provided in the text. Antenna design and feeds, type of receivers, absolute calibration, correction for atmospheric absorption variations and possibly others should be added.

Reply: Because the main goal of this article is to study the signals of non-thermal energy release in the solar transition region, the whole paper is organized around this goal, including the proposed ultra-wide broadband millimeter-wave spectrometers for detecting the aforementioned signals. So, our focus is on discussing the possible characteristics of the non-thermal emission in the solar transition region and on revealing the energy release and mass propagations from the chromosphere and lower solar atmosphere to the corona. As these signals should be many, short-lived, narrow bandwidth, fast frequency drifting and very weak, during the observation experiments, we will attempt to change the parameters of the telescope, including increasing the number of antenna elements to improve the sensitivity of the observation and discover these small-scale energy release events, so we have not presented enough information about the proposed instruments.

  1. As a full-disk instrument will only be capable of detecting only flare-associated emission, most of the discussion in Sections 1-2 is not very relevant.

Reply: As our discussion above, this instrument is proposed mainly to detect the signals of non-thermal energy release in the solar transition region, therefore, it mainly observe the quiet Sun without solar flares. Of course, as the test-SUBMS will begin the test observation in the autumn of 2024, around that time, the Sun will come into its peak year of the solar cycle 25, and the test-SUBMS can also observe the flare-associated millimeter-wave bursts of high-energy processes in the corona.

I suggest that the authors completely re-organize their text. If they wish to build their text around the spectrometer proposal they should follow the above suggestions and find a more appropriate title. If, on the other hand, they wish to write about the radio emission from the transition region, they should consider a more extended review article with many more details and references.

Reply: as we mention above, the main goal of this paper is to discuss the possible signals of non-thermal energy release in the solar transition region, the whole paper is organized around this goal, including the proposed millimeter-wave spectrometers for detecting the aforementioned signals. In our revised version, we rewrite the whole manuscript, add 24 references.

Concerning ALMA, the authors may wish to consult the articles https://doi.org/10.3389/fspas.2022.977368, https://doi.org/10.3389/fspas.2022.966444, https://doi.org/10.3389/fspas.2022.981205 and https://doi.org/10.1051/0004-6361/202244532.

Reply: in the revised version, we added these references.

Reviewer 4 Report

Comments and Suggestions for Authors

The manuscript is devoted to discussion of the spectral characteristics of millimeter-wave emission from the transition region and proposing a new conception of ultra-wide broadband millimeter-wave spectrometers to observe them. For this, authors proposing to construct experimental ground-based broadband millimeter-wave spectrometers in frequency range of 20-35 GHz or 75-100 GHz at a place of Saisteng Mountain in Qinghai province, China and launching a telescope beyond the earth atmosphere to obtain the broadband millimeter-wave dynamic spectrum in the full frequency range. The manuscript focused on the current status of radio observations of the solar transition region and the planned observation system will have possibility to make a great contribution to well understanding of the solar transition region in future.

In general, the manuscript is well written and has a good quality, but there are some points that I listed, typos and nonexistence of year in some of the references need to correct before acceptance. The abstract is an objective representation of the manuscript and summarizes the paper. The Introduction section is comprehensible, well-structured and presents the basis and motivation of the study. Section 2 discusses the radio emission mechanisms in the solar transition region with some critical questions with their answers including citations to the literature. History and current status of transition zone radio observations at millimeter wavelength and the new conception of solar ultra-wide broadband millimeter-wave spectrometers are presented with detail as two new sections.  The section of conclusion clearly pointed out the proposed new systems. The quality of plots and tables are good.

When I looked the literature I found many recent papers about the topic, but in the manuscript there are only two papers cited since 2022. Therefore I strongly recommend adding some recent studies i.e. X Zhang, L Deng et al 2023 ApJ, DE Gary, Annual Review of Astronomy and Astrophysics, 2023, CE Alissandrakis, ASR, to 2023, etc. to the manuscript and update the manuscript correspondingly.

Line 142-144. Author mentioned that “There are many small-scale explosive phenomena, such as micro-flares, nano-flares, spicules, bright points, and even the prevalence of small-scale jets from the networks of the transition region, which are associated with magnetic reconnection [17, 4].” All these phenomena mainly occur in the solar chromosphere and they called as a chromospheric fine structures. But in the manuscript they mentioned as transition region phenomena and there is no any information about their connection with the solar chromosphere. I think the source region of these small scale events should be presented with more detail.

General comment:  Please avoid to write “etc.” in the manuscript. It is essential to provide accurate information.

Line 20. “…magnetic fields and variations.”  “…magnetic fields and their variations.”

Line 104. “Kaufmann et al. reported”, please add year

Line 181. “energy spectral”, spectral energy

Line 240. Please add information about green color to the caption of Figure 2.

Comments on the Quality of English Language

I already add my comments above

Author Response

Reply to the Review report 2:

The manuscript is devoted to discussion of the spectral characteristics of millimeter-wave emission from the transition region and proposing a new conception of ultra-wide broadband millimeter-wave spectrometers to observe them. For this, authors proposing to construct experimental ground-based broadband millimeter-wave spectrometers in frequency range of 20-35 GHz or 75-100 GHz at a place of Saisteng Mountain in Qinghai province, China and launching a telescope beyond the earth atmosphere to obtain the broadband millimeter-wave dynamic spectrum in the full frequency range. The manuscript focused on the current status of radio observations of the solar transition region and the planned observation system will have possibility to make a great contribution to well understanding of the solar transition region in future.

In general, the manuscript is well written and has a good quality, but there are some points that I listed, typos and nonexistence of year in some of the references need to correct before acceptance. The abstract is an objective representation of the manuscript and summarizes the paper. The Introduction section is comprehensible, well-structured and presents the basis and motivation of the study. Section 2 discusses the radio emission mechanisms in the solar transition region with some critical questions with their answers including citations to the literature. History and current status of transition zone radio observations at millimeter wavelength and the new conception of solar ultra-wide broadband millimeter-wave spectrometers are presented with detail as two new sections.  The section of conclusion clearly pointed out the proposed new systems. The quality of plots and tables are good.

Reply: Firstly, the authors greatly appreciate the reviewer's affirmation and helpful suggestions for this article. We have carefully and meticulously revised the entire text based on the comments of the reviewers, rewriting most of the paragraphs and correcting previous inaccurate descriptions. Now, the entire text should appear more reasonable and complete. In the following, we will answer the questions raised by the reviewers one by one.

When I looked the literature I found many recent papers about the topic, but in the manuscript there are only two papers cited since 2022. Therefore I strongly recommend adding some recent studies i.e. X Zhang, L Deng et al 2023 ApJ, DE Gary, Annual Review of Astronomy and Astrophysics, 2023, CE Alissandrakis, ASR, to 2023, etc. to the manuscript and update the manuscript correspondingly.

Reply: In the revised manuscript, we have added 17 references, including the latest related publications from recent years and the references mentioned by the reviewer.

Line 142-144. Author mentioned that “There are many small-scale explosive phenomena, such as micro-flares, nano-flares, spicules, bright points, and even the prevalence of small-scale jets from the networks of the transition region, which are associated with magnetic reconnection [17, 4].” All these phenomena mainly occur in the solar chromosphere and they called as a chromospheric fine structures. But in the manuscript they mentioned as transition region phenomena and there is no any information about their connection with the solar chromosphere. I think the source region of these small scale events should be presented with more detail.

Reply: Yes. Many people think that the small-scale explosive phenomena, such as micro-flares, nano-flares, spicules, bright points, and even small-scale jets mainly occur in the solar chromosphere and they called as a chromospheric fine structures. However, there are also many observations showed that many of these small-scale explosive events belong to transition region structures. For example, based on observations of several rocket flights, Porter & Dere (1991) reported many small-scale and short-lived (typically <1 minute) jets and turbulent events with non-Gaussian enhancements in both red and blue wings of transition region lines covering temperature range from K to K. Tian et al. (2014) applied the UV observations from the satellite IRIS and found that most of these small-scale events have temperature of at least  K and constitute an important elements of the transition region structures, and proposed that these events are associated with magnetic reconnections. In our revised version, we rewrite this paragraph, accordingly.

General comment:  Please avoid to write “etc.” in the manuscript. It is essential to provide accurate information.

Reply: In the revised draft, we have specifically revised these expressions.

Line 20. “…magnetic fields and variations.”  “…magnetic fields and their variations.”

Reply: We correct it in the revised version.

Line 104. “Kaufmann et al. reported”, please add year

Reply: We correct it in the revised version.

Line 181. “energy spectral”, spectral energy

Reply: We correct it in the revised version.

Line 240. Please add information about green color to the caption of Figure 2.

Reply: We have added the information to the caption of Figure 2 in the revised version “The green characters represent the upper and lower limits of the frequency range and the corresponding wavelengths of SUBMS.”

Round 2

Reviewer 2 Report

Comments and Suggestions for Authors

I have analyzed the revised version of manuscript universe-2790776 by Tan et al. . I recognize author's efforts to increase the quality of the paper. Most of my questions/concerns on the original submission were answered adequately. I consider the paper deserves publication after minor following revisions.  

1) for my comment 2) the authors responded that in the transition region BETA "plasma is still much smaller than 1". The range in density and temperature do not appear in their response. Please indicate these ranges, and the corresponding value for BETA. I just remember that, even with BETA < 1, for example BETA = 0.2, the acceleration of electrons is seriously affected.  

2) for my comment 3) on magnetic reconnection, authors do not answer on the number of accelerated electrons. This number is important for the level of flux density authors pretend to measure in radio waves  

3) for my comment 5: during flare and during non-flare period, emission will be dominated by the thermal background radiation of the solar disk. So that the proposed new instrument will observe (very) small excesses relative to the background. It is a serious instrumental matter that authors need to take into account for the instrument definition and characteristics.   

other comment:  

line 202: this opacity formula is only valid for fully ionized plasma. At the base of the transition region this may not be the case. Therefore, breemsstrahlung opacity may include the contribution of H- (see e.g De la Luz et al. 2011, ApJ, 737)

Author Response

Second reply to review report 2

I have analyzed the revised version of manuscript universe-2790776 by Tan et al. I recognize author's efforts to increase the quality of the paper. Most of my questions/concerns on the original submission were answered adequately. I consider the paper deserves publication after minor following revisions.  

Reply: Firstly, the authors greatly appreciate the reviewer's affirmation and helpful suggestions for this article. We have carefully and meticulously re-revised the entire text based on the review report, rewriting most of the paragraphs and correcting previous inaccurate descriptions. In the following, we will answer the questions raised by the reviewer one by one.

1) for my comment 2) the authors responded that in the transition region BETA "plasma is still much smaller than 1". The range in density and temperature do not appear in their response. Please indicate these ranges, and the corresponding value for BETA. I just remember that, even with BETA < 1, for example BETA = 0.2, the acceleration of electrons is seriously affected.  

Reply: In the third paragraph of Section 2, we presented the estimation of these parametric ranges “We know that compared to the most flaring source region mainly locating in the corona, the solar transition region has higher density (    to   , see Fig. 1), stronger magnetic field (from about 100 G to nearly 2000 G, corresponding from the solar quiet region, networks to the sunspot active region), and lower temperature (from K to  K, also see Fig.1). With these parameters, we may calculate the plasmas  is in the range from  around the top of transition region above sunspot active regions to 0.01 near the bottom of the transition region above the solar quiet region. That is to say, the plasmas still have . In fact, except the transition region above areas where the magnetic field is too weak (for example, B < 100 G), the solar transition region above the sunspot regions, network structures, basically meet the conditions of collisionless magnetized plasma.”

2) for my comment 3) on magnetic reconnection, authors do not answer on the number of accelerated electrons. This number is important for the level of flux density authors pretend to measure in radio waves  

Reply: We added the estimation of the possible number of accelerated electrons from each small-scale reconnecting current-sheet and the total number from all small-scale reconnecting current-sheets associated with the transition region above the whole solar surface in the section 2.1 of the revised version. Our estimation shows the total number of accelerated electrons is about  for each reconnected current sheet in the transition region, and may produce a radio emission flux of 0.2-2.0 sfu at frequency of 35 GHz. We referred such non-thermal radio emission as transition region small-scale microwave bursts (TR-SMBs), which is weak, narrowband, short lifetime, rapid frequency drifting, and in great number. Comparing to the quiet Sun flux intensity of about 2400 sfu, it is required a sensitivity better than 0.01-0.1% and the dynamic range of the telescope should be higher than 30 - 40 dB. The total number is about  accelerated electrons and the total number of TR-SMBs is about  per second in the transition region above the magnetic network on the whole solar surface. The main aim of the proposed telescope is to observe these TR-SMBs.

3) for my comment 5: during flare and during non-flare period, emission will be dominated by the thermal background radiation of the solar disk. So that the proposed new instrument will observe (very) small excesses relative to the background. It is a serious instrumental matter that authors need to take into account for the instrument definition and characteristics.   

Reply: In our revised version, we added the estimation of the possible radio flux intensity radiated by the non-thermal electrons from the transition region. The results show that the emission flux intensity is in the range of 0.2 - 2 solar flux unit (sfu) at frequency of 35 GHz. Comparing to the quiet Sun flux intensity of about 2400 sfu, it is required a sensitivity better than 0.01-0.1% and the dynamic range of the telescope should be higher than 30 - 40 dB for the new instrument. In Table 3, we required that the instrument should have sensitivity of <0.2 sfu while the dynamic range >30 dB.

other comment:  

line 202: this opacity formula is only valid for fully ionized plasma. At the base of the transition region this may not be the case. Therefore, bremsstrahlung opacity may include the contribution of H- (see e.g De la Luz et al. 2011, ApJ, 737)

Reply: In the revised version, we rewrote this part. According to the results of De la Luz et al. (2011), the contribution of H- process is dominant below 500 km above the photosphere, and bremsstrahlung processes are dominant mainly in chromosphere below the bottom of the transition region. However, in the transition region, temperatures exceed 20000 K, the non-ionized neutral particles and their contributions to the radio emission can be almost negligible. Especially for our main target (TR-SMBs) in relatively strong magnetic field regions (including sunspot regions, network regions), the emission is dominated by gyrosynchrotron processes, and bremsstrahlung emission almost can be ignored.

Reviewer 3 Report

Comments and Suggestions for Authors

Report on the revised version of "Radio emission and observations from the solar transition region" by Tan et al.

The authors have made changes in response to the reviewers' comments. However, the revised manuscript is still below my expectations. In particular:

1. I insist that the authors provide more characteristics of the proposed instrument, as I asked in my first report. For example, what kind of spectral analyzer is required to achieve the sensitivity of < 0.1 sfu, as stated in Table 3? (at 35 GHz, with the quiet sun flux of ~ 2400 sfu, this requires a gain stability better 0.004% and a dynamic range > 40 db). What is the expected  level and the time scale of the atmospheric transparency fluctuations and how they will be corrected?

2. From the authors' reply and the revised manuscript I understand that they want to look for weak transient emissions. It appears to me that the proposed instrument, or any full-disk instrument for that matter, will only be able to detect flare associated emissions and their fluctuations, as smaller scale transients are too weak and too many. This is the reason that I mentioned that "most of the discussion in Sections 1-2 is not very relevant". I will not insist on that comment, but the authors must provide some numbers on the expected flux level and repetition rate, both for thermal emission (based, e.g., on the review on ALMA transients that I suggested in my first report) and for non-thermal emission, and compare these numbers with the characteristics of their proposed instrument. Any insight from CBS?

3. I do insist that the authors find a more suitable title. The present title suggests a review article and this is not one.

- Some more comments and corrections:

l. 41: About the height of the TR see your reference [1]

l. 42:  "collision chromoshere" -> "collisioninal chromoshere"

l. 43:  "non-collision corona" -> "collisionless corona

l. 55: Start a new paragraph with "We know...

l. 61:  "solar winds" -> "the solar wind"

l. 75:  "many information" -> "much information"

l. 91:  "Besides the dominance of the emission mechanism" -> "Regardless of the emission mechanism"

l. 92:  "solar surface" -> "photosphere"

l. 94:  "For example, the radio emission at centimeter and decimeter wavelengths produces in the lower corona and always associated with the source regions of solar flares, while the radio emission at meter and decameter wavelengths generates from the higher corona and always related to the propagations of flaring non-thermal electron beams, CMEs and the related shock waves" -> "For example, flare-associate radio emission at centimeter and decimeter wavelengths is produced in the lower corona near the source regions of solar flares, while the radio emission at meter and decameter wavelengths is generated in the higher corona and always related to the propagation of non-thermal electron beams, CMEs and the related shock waves"

l. 113-115: The emission frequency at short wavelengths is not related to the plasma frequency, and the formation height is determined by the opacity (see, e.g., Fig. 2 in https://iopscience.iop.org/article/10.1086/523671)

l. 123: "contributed" -> "attributed"

l. 146: "there must" -> "we must"

l. 351: "there has already CBS which" -> "CBS already"

Comments on the Quality of English Language

see above

Author Response

Second reply to review report 3

Report on the revised version of "Radio emission and observations from the solar transition region" by Tan et al.

Reply: Firstly, the authors greatly appreciate the reviewer's affirmation and helpful suggestions for this article. We have carefully and meticulously re-revised the entire text based on the review report, rewriting most of the paragraphs and correcting previous inaccurate descriptions. In the following, we will answer the questions raised by the reviewer one by one.

The authors have made changes in response to the reviewers' comments. However, the revised manuscript is still below my expectations. In particular:

  1. I insist that the authors provide more characteristics of the proposed instrument, as I asked in my first report. For example, what kind of spectral analyzer is required to achieve the sensitivity of < 0.1 sfu, as stated in Table 3? (at 35 GHz, with the quiet sun flux of ~ 2400 sfu, this requires a gain stability better 0.004% and a dynamic range > 40 db). What is the expected level and the time scale of the atmospheric transparency fluctuations and how they will be corrected?

Reply: in the revised version, we have supplemented and improved Tables 2 and 3 for the characteristics of the proposed telescope, including antenna and feed design, as well as calibration methods. Our observational targets are very numerous and very weak transition region small-scale microwave bursts (TR-SMBs). According to our estimation, their emission flux intensity is in the range of 0.2 - 2 solar flux unit (sfu) at frequency of 35 GHz, only 0.01-0.1% of the quiet Sun flux intensity, lifetime is in ms- timescale, with rapid frequency drifting rates, the number is about  TR-SMBs per second above the full solar disk. These characteristics require the proposed telescope to have high temporal resolution (<0.2 ms), high frequency resolution (<200 MHz), high sensitivity (<0.2 sfu) and large dynamic range (>30 dB). In order to achieve the above goals, we first recommend deploying the telescope (SUBMS) to the Chinese Space Station, which can avoid the impact of absorption and transparency fluctuations of the Earth's atmosphere. We plan to select front-fed parabolic reflector antenna and quadruple riddged flared horn as the feed for frequency of 20-40 GHz and 40-60 GHz, Cassegrain dual-reflector antenna and Corrugated Horn feed for frequency of 60-80 GHz and 80-100 GHz. For the ground-based test-SUBMS (20-35 GHz), besides the antenna and feed design, we choose to install the telescope on the Saisteng Mountain in Qinghai Plateau, China, where the air is very clean and very dry, where can minimize the impact of atmospheric absorption. Additionally, we also consider the corrections of atmospheric absorption and transparency fluctuations by using atmospheric absorption model. As our observational targets are very numerous, millisecond-lifetime, and the fluctuation of the environmental temperature is always at minute or even longer timescale, it is enough to ignore the atmospheric transparency fluctuations. However, we still plan to install a temperature-controlling system in the receiving system to minimize the impact of environmental temperature changes on observations.

  1. From the authors' reply and the revised manuscript I understand that they want to look for weak transient emissions. It appears to me that the proposed instrument, or any full-disk instrument for that matter, will only be able to detect flare associated emissions and their fluctuations, as smaller scale transients are too weak and too many. This is the reason that I mentioned that "most of the discussion in Sections 1-2 is not very relevant". I will not insist on that comment, but the authors must provide some numbers on the expected flux level and repetition rate, both for thermal emission (based, e.g., on the review on ALMA transients that I suggested in my first report) and for non-thermal emission, and compare these numbers with the characteristics of their proposed instrument. Any insight from CBS?

Reply: Yes, the main purpose of this article is to discuss the signals of non-thermal processes occurring in the solar transition region and how to observe them. We referred these signals as TR-SMBs. In the revised version, we present the estimation related to TR-SMBs, the number of the related non-thermal electrons is about , and the emission flux intensity is in the range of 0.2 - 2 solar flux unit (sfu) at frequency of 35 GHz, only 0.01-0.1% of the quiet Sun flux intensity. In fact, signals of RS-SMBs are the components of the background quiet Sun flux intensity in the centimeter-millimeter wavelengths, equivalent to the fluctuations on the background of the thermal emission. Their lifetime is in ms- timescale, with rapid frequency drifting rates, the number is about  TR-SMBs per second above the full solar disk. In order to detect the above weak, narrowband, short lifetime, rapid frequency drifting, and so much TR-SMBs, we require the proposed telescope to have high temporal resolution (<0.2 ms), high frequency resolution (<200 MHz), high sensitivity (<0.2 sfu) and large dynamic range (>30 dB), so that the signal will not be averaged and smoothed out. Such a proposed telescope is different from CBS which is dedicated to observe the bursts associated to solar flares, but it can also be used to observe solar flare bursts. CBS has temporal resolution of 134 ms and frequency resolution of 153 kHz which may be integrated into 500 MHz. It can obtain the clear profiles and spectrum of solar flares. However, its frequency range is only from 35 GHz to 40 GHz, and the bandwidth is 5 GHz, which is not enough wide to cover the wavelengths of non-thermal process in the transition region, and the temporal resolution (134 ms) is not enough short to discriminate the TR-SMBs. Therefore, CBS can be used to observe solar flares, but not suitable to observe the signals of TR-SMBs. In fact, we need a much more wider broadband centimeter-millimeter-wave dynamic spectrometer with higher temporal resolution in the frequency range from 20 GHz to 100 GHz.

  1. I do insist that the authors find a more suitable title. The present title suggests a review article and this is not one.

Reply: In the revised version, we try to change the title into: “Non-thermal radio emission of the solar transition region and a proposed observational program”. And we hope that is more suitable for this article.

- Some more comments and corrections:

  1. 41: About the height of the TR see your reference [1]

Reply: Here, we change the expression as “the height of a few thousands km (from 0.14 Mm to 5.7 Mm, depending on observations at different wavelengths [1])”

  1. 42:  "collision chromosphere" -> "collisional chromosphere"

Reply: OK. We do this correction in the revised version.

  1. 43:  "non-collision corona" -> "collisionless corona

Reply: We do this correction in the revised version.

  1. 55: Start a new paragraph with "We know...

Reply: We do this correction in the revised version.

  1. 61:  "solar winds" -> "the solar wind"

Reply: We do this correction in the revised version.

  1. 75:  "many information" -> "much information"

Reply: We do this correction in the revised version.

  1. 91:  "Besides the dominance of the emission mechanism" -> "Regardless of the emission mechanism"

Reply: We do this correction in the revised version.

  1. 92:  "solar surface" -> "photosphere"

Reply: We do this correction in the revised version.

  1. 94:  "For example, the radio emission at centimeter and decimeter wavelengths produces in the lower corona and always associated with the source regions of solar flares, while the radio emission at meter and decameter wavelengths generates from the higher corona and always related to the propagations of flaring non-thermal electron beams, CMEs and the related shock waves" -> "For example, flare-associate radio emission at centimeter and decimeter wavelengths is produced in the lower corona near the source regions of solar flares, while the radio emission at meter and decameter wavelengths is generated in the higher corona and always related to the propagation of non-thermal electron beams, CMEs and the related shock waves"

Reply: We do this correction in the revised version.

  1. 113-115: The emission frequency at short wavelengths is not related to the plasma frequency, and the formation height is determined by the opacity (see, e.g., Fig. 2 in https://iopscience.iop.org/article/10.1086/523671) (Avrett & Loeser, ApJS, 2008)

Reply: In the revised version, we considered and rewrote this paragraph and added the related reference.

  1. 123: "contributed" -> "attributed"

Reply: We correct it in the revised version.

  1. 146: "there must" -> "we must"

Reply: In the revised version, we rewrite this sentence as: “Now that the solar transition region is highly dynamic, there should be frequently non-thermal energy release and conversion occurring here, resulting in the generation of non-thermal electrons.”

  1. 351: "there has already CBS which" -> "CBS already"

Reply: We rewrite it in the revised version following the above suggestion.
